# Targeting Adenosine Receptor Signaling in Cancer Immunotherapy

**DOI:** 10.3390/ijms19123837

**Published:** 2018-12-02

**Authors:** Kevin Sek, Christina Mølck, Gregory D. Stewart, Lev Kats, Phillip K. Darcy, Paul A. Beavis

**Affiliations:** 1Cancer Immunology Program, Peter MacCallum Cancer Centre, East Melbourne, Victoria 3000, Australia; Kevin.Sek@petermac.org (K.S.); lev.kats@petermac.org (L.K.); 2Sir Peter MacCallum Department of Oncology, The University of Melbourne, 3010 Parkville, Australia; 3Department of Pathology, University of Melbourne, Parkville 3010, Australia; christina.moelck@gmail.com; 4Drug Discovery Biology, Monash Institute of Pharmaceutical Sciences and Department of Pharmacology, Monash University, Parkville 3052, Australia; gregory.stewart@monash.edu; 5Department of Immunology, Monash University, Clayton 3052, Australia

**Keywords:** Adenosine, Adenosine receptors, immune cells, tumor cells, cancer immunotherapy

## Abstract

The immune system plays a major role in the surveillance and control of malignant cells, with the presence of tumor infiltrating lymphocytes (TILs) correlating with better patient prognosis in multiple tumor types. The development of ‘checkpoint blockade’ and adoptive cellular therapy has revolutionized the landscape of cancer treatment and highlights the potential of utilizing the patient’s own immune system to eradicate cancer. One mechanism of tumor-mediated immunosuppression that has gained attention as a potential therapeutic target is the purinergic signaling axis, whereby the production of the purine nucleoside adenosine in the tumor microenvironment can potently suppress T and NK cell function. The production of extracellular adenosine is mediated by the cell surface ectoenzymes CD73, CD39, and CD38 and therapeutic agents have been developed to target these as well as the downstream adenosine receptors (A_1_R, A_2A_R, A_2B_R, A_3_R) to enhance anti-tumor immune responses. This review will discuss the role of adenosine and adenosine receptor signaling in tumor and immune cells with a focus on their cell-specific function and their potential as targets in cancer immunotherapy.

## 1. Introduction

Adenosine triphosphate (ATP) is a ubiquitous molecule that plays a vital role as the universal energy currency within the cell. Under physiological conditions, intracellular ATP concentrations are maintained at millimolar concentrations, while extracellular levels are tightly regulated in the nanomolar range [1,2]. However, under certain conditions, such as tissue injury, inflammation, ischemia, or in the tumor microenvironment (TME), extracellular ATP levels increase due to release from inflammatory, apoptotic, or necrotic cells [3]. Extracellular ATP signals through P2 receptors (P2R) that are widely expressed on immune and non-immune cells within the body and are involved in multiple physiological and pathological processes. The current paradigm of purinergic signaling on the immune response can be described as a balance between pro- and anti- inflammatory signaling from extracellular ATP and adenosine (ADO), respectively. Physiologically, ATP released from stressed, apoptotic, and necrotic cells can act as a ‘danger signal’ during the acute inflammatory response and is essential for the clearance of intracellular bacteria, parasites, and viruses [4]. ATP can also induce a form of immunogenic cell death in cancer cells that promotes immunosurveillance in the TME (reviewed in [5]). In contrast, ADO is mainly anti-inflammatory and promotes cytoprotection [6], wound healing [7], and suppression of the immune system. Whilst the concentration of ADO in normal tissue resides around nanomolar concentrations, it has been shown to be present at up to micromolar concentrations in solid tumors and enriched in the hypoxic tumor core [2,8,9]. Increased ADO levels are furthermore observed in inflammation, ischemia, hypoxia, and organ trauma, and is a major component in the regulation of immune cells in the context of bacterial/viral sepsis or renal dysfunction or injury (reviewed in [10,11]). The critical role for ADO signaling in immune regulation is further emphasized by the total dysfunction of T cells, NK cells, and B cells in individuals with a variant of severe combined immunodeficiency (SCID) as a result of mutations in adenosine deaminase (ADA) that catalyzes the conversion of ADO to inosine [12].

There are four known subtypes of ADO receptors (A_1_R, A_2A_R, A_2B_R, A_3_R) which have distinct expression patterns and mediate diverse signaling pathways. Due to the presence of high concentrations of ADO within the TME and the expression of ADO receptors on tumor and immune cells, the role of ADO in cancer progression and anti-tumor immune responses have been intensively investigated. This has led to the clinical development of antibodies and small molecule inhibitors targeting various components of the ADO pathway including CD39, CD38, CD73, A_2A_R, and A_2B_R. Despite this, the mechanisms of action of these reagents in terms of their target cell population and intracellular signaling pathways remain relatively unknown. This review will discuss the signaling pathways in which ADO receptors mediate their effect in both tumor and immune cells, and recent progress in targeting the ADO pathway to improve immunotherapies.

## 2. Extracellular Adenosine Production in the Tumor Microenvironment

The TME exhibits high concentrations of ADO due to the contribution of immune and stromal cells, tissue disruption, and inflammation. A predominant driver is hypoxia due to the lack of perfusion that can lead to cellular stress [13,14], and secretion of large amounts of ATP (reviewed in [15]). Hypoxia also drives expression of the well-defined transcription factor HIF1α, which promotes the expression of ectoenzymes CD39 (NTPDase1) and CD73 (5’-NT) on tumor cells, stromal cells, and tumor infiltrating immunosuppressive cell subsets such as regulatory T cells (Treg) and myeloid derived suppressor cells (MDSC) [16,17]. CD39 catalyzes the conversion of ATP and ADP into AMP, while CD73 catalyzes the irreversible conversion of AMP into ADO [18] (Figure 1). Supporting their role in inflammation and tumorigenesis, mice deficient in CD39 or CD73 are susceptible to inflammation/autoimmunity and are resistant to tumor growth due to alleviation of ADO-mediated immunosuppression [19,20,21]. Furthermore, CD39 and CD73 have been shown to be biomarkers of patient outcomes in several tumor types, with the majority of studies linking high expression to poorer clinical outcomes in triple negative breast, lung, ovarian, kidney, gastric cancer, and melanoma. However, other studies have also linked high expression of CD73 with positive outcomes in certain cancers such as bladder or colorectal cancers [22,23,24,25,26,27,28]. The reasons for these discrepancies are not fully understood but are potentially related to the relative contribution of the anti-tumor immune response in each cancer type. Tumors that have lower expression of ectonucleotidases may have increased extracellular ATP levels that can play a dual role in the TME by acting on P2 receptors to fine-tune the immune response. ATP can drive the recruitment and activation of inflammatory cells, particularly antigen presenting cells, and increase their capacity to perform antigen presentation but has also been shown to attract Treg cells and promote T-helper 2 (T_H_2) or Treg cell differentiation which may instead promote an immunosuppressive TME (reviewed in [15]). Although AMP production is thought to be predominantly mediated by CD39, an alternative source of AMP is the conversion of NAD^+^ by CD38 and CD203a receptors [29,30,31,32] and through the activity of tissue non-specific alkaline phosphatases [33,34]. CD38 is expressed on tumors, T cells, and NK cells and can promote ADO generation and subsequent suppression of T cell function and proliferation [35,36]. Finally, ADO is rapidly removed from the extracellular space by conversion into inosine (INO) by ADO deaminase (ADA) or uptake by nucleoside transporters (NT) back into the cell, whereby it is converted back into AMP by ADO kinases [37]. Thus, ADO accumulates in the TME predominantly through the catabolism of extracellular ATP to ADO by CD73, CD39, and CD38 expressed on tumor and immune cells, and has been reported to mediate suppression of anti-tumor immunity through activation of ADO receptors (Figure 1).

## 3. The Diversity of Adenosine Receptor Signaling

The four known subtypes of ADO receptors (A_1_R, A_2A_R, A_2B_R, and A_3_R), all of which are G-protein coupled receptors (GPCRs), have a high degree of homology between human, mouse, and rat orthologues. Protein homology ranges between 85% to 90% for each receptor subtype except the A_3_R, which shows the greatest variability between species with only a 78.9% homology between mouse and human [38]. Endogenous ADO has a potency for the A_1_R, A_2A_R, and A_3_R in the high nM range. The A_2B_R however is considered a relatively low affinity receptor with a potency in the micromolar range [39,40]. Interestingly, the human A_3_R but not the rat A_3_R has higher affinity for ADO comparably, with K_i_ of 290 and 6500 nM respectively, which suggests a potential divergence of function between species [39]. Thus, at physiological concentrations of ADO, signaling is primarily mediated via the A_2A_R, A_1_R, and A_3_R with the A_2B_R only being activated during elevated levels of ADO under pathophysiological conditions such as in the inflammatory TME. Although ADO is the predominant ligand for the ADO receptors, INO has also been identified as a possible partial agonist for the A_2A_R and A_3_R indicating that the role of non-canonical ligands for the ADO receptors should also be considered [41,42,43,44]. Similar to other GPCRs, ADO receptors can mediate ligand-specific signaling events known as biased agonism [45]. For example, activation of the A_2A_R by INO preferentially promotes extracellular signal-regulated kinases 1 and 2 (ERK1/2) signaling, whereas ADO induced signaling is biased towards cAMP [46]. As many of the ADO receptors are co-expressed on the same cell, their signaling pathways can be complex with multiple mechanisms to consider, such as the interplay between regulation of expression, trafficking, and desensitization of ADO receptors [47,48], ligand–receptor interactions (reviewed in [49], receptor–receptor interactions (reviewed in [39]), and the spatial and temporal compartmentalization of signaling (reviewed in [50,51]). While these mechanisms undoubtedly play a major role in immune cell regulation and could explain the diversity of cellular responses, they are beyond the scope of this review, and here we will focus on downstream pathways of ADO signaling characterized by individual cell types and their effect, with a focus on tumor cells and tumor-infiltrating immune cell subsets.

GPCRs couple with intracellular signal transducers such as the heterotrimeric G-proteins, which consists of the Gα, Gβ, and Gγ subunits. While the Gα subunit can signal independently, the Gβ and Gγ subunits can only signal as obligate dimers (Gβγ). The Gα subunit can be classified into 4 major families (Gαs, Gαi/o, Gq/11, Gα12/13) and is a primary signaling modality of the A_2A_R, A_2B_R and the A_1_R, A_3_R receptors which couple to the Gαs and Gαi/o family of Gα proteins respectively. In addition to these, the A_2A_R also has the capacity to couple to the Golf that has expression limited to the brain, while the A_2B_R [52] and the A_3_R can also bind to the Gq/11 proteins that promote phospholipase C (PLC) dependent intracellular Ca^2+^ signaling [40,53,54]. Stimulation of adenylate cyclase (AC) by Gαs proteins induced by stimulation of A_2A_R/A_2B_R in T cells results in the localized accumulation of cyclic AMP (cAMP) in microdomains that occur at the immunological synapse (IS) (reviewed in [55]). cAMP can bind to the regulatory domains on type 1 Protein Kinase A (PKA) targeted to lipid rafts in the IS by Ezrin, an A-kinase binding protein (AKAP) [55]. This results in the phosphorylation of Src kinase, a negative regulator of Lck kinase, a key mediator of T-cell receptor (TCR) signaling activity [56,57,58,59]. Other putative cAMP mediators of T cell suppression include EPAC1, that is involved in the direct transfer of cAMP from Treg to effector T cells (Teff) through gap junctions [60,61], and rap1 that suppresses Teff function and is a known downstream pathway of the checkpoint receptor CTLA-4 [62,63]. In addition to suppression of T cell responses, cAMP and PKA have an overall inhibitory effect on other immune cells including B cells, neutrophils [64,65,66], monocytes/macrophages [67,68], and NK cells [69]. Whilst the A_2A_R and A_2B_R increase cAMP concentrations, leading to immune cell suppression, the A_1_R and A_3_R limit cAMP accumulation through Gαi mediated suppression of AC activity. The A_1_R and A_3_R can therefore potentially activate opposing signaling pathways in immune cells, exemplified by A_1_R and A_3_R specific agonists reversing inhibition of cytokine production and proliferation by the A_2A_R in a mixed lymphocyte culture [70]. A_3_R stimulation has also been shown to suppress cAMP levels in T cells [71,72]. Notably, these studies were performed exclusively with pharmacological agents that are specific for the respective receptors but may still have some activity on the other receptor subtypes and so do not preclude the possibility that these signaling events are exclusively A_1_R and A_3_R mediated. Aside from the Gα subunit which signals independently, the Gβ and Gγ subunits can also signal as obligate heterodimers. In addition, β-Arrestins or AKAPs can interact with the ADO receptors as scaffolding proteins to recruit a host of other signaling proteins that form the signalosome [49]. These proteins include, but are not limited, to mitogen-activated protein kinases (MAPK), PLC, Protein kinase C (PKC), Protein kinase D, RhoA, ERK, PI3K/AKT, or the mTOR proteins [53,73,74,75,76,77,78,79,80]. While a myriad of downstream pathways of ADO receptor signaling have been identified and suggested to play a role in tumor and immune cell function, many of the signaling pathways are cell type dependent and only a few studies have actually shown a direct link between signaling pathways and phenotypic observations in primary immune cells. Given that these signaling events are determined by a complex interplay governed by the cellular expression of the different ADO receptors as well as the proteins involved in the downstream signaling pathways, it is important to consider these signaling pathways in a cell type specific context.

## 4. Expression of Adenosine Receptors and Signaling Pathways in Tumor Cells

ADO receptors are expressed in hematopoietic and non-hematopoietic tissues and therefore are often expressed in both hematological and non-hematological cancers. In addition, there are several reports of tumor expression of all four ADO receptors: A_1_R [71,81], A_2A_R [28,71,81,82,83], A_2B_R [84,85], and A_3_R [71,73,86,87]. Due to the diverse expression of ADO receptors on distinct tumor types (and likely differential signaling induced) it is perhaps unsurprising that a range of phenotypes have been observed following stimulation of tumor cells by ADO. These include diverse effects on proliferation, apoptosis, cytoprotection, and migration (reviewed by [88]). The most extensively studied have been the role of the A_3_R and A_2B_R on cancer cells. A_3_R has been reported to be upregulated in primary and metastatic tumors (relative to healthy matched tissue) in colorectal cancer and breast cancer [89,90] and some studies have suggested that A_3_R blockade could be a novel oncology target due to its higher expression on cancer tissues. On the other hand, the ability of A_3_R activation to reduce tumor proliferation and growth has been shown by numerous groups using tumors derived from sarcomas, melanomas, lymphomas, lung tumors, and prostate cancers [88,91,92,93,94,95,96]. Indeed, the A_3_R agonists CL-IBMECA and CF102 have shown promise in vitro by inducing apoptosis in multiple tumor types through the suppression of PKA, ERK, and AKT pathways [80,97,98,99]. Conversely, it has also been reported that inosine-mediated activation of the A_3_R can enhance melanoma cell proliferation through activation of the ERK pathway [100] and a cytoprotective role for the A_3_R has also been described [101]. Similarly, activation of the A_3_R has been suggested to enhance the migration or invasion of tumor cells [73,102,103]. Notably, in one study, divergent effects of A_3_R activation (via IB-MECA stimulation) on proliferation were observed in two colorectal cancer lines. A_3_R activation was shown to decrease the proliferation and survival of HCT-116 cells through the protein phosphatase 2A pathway, whereas the proliferation and survival of HT-29 cells was shown to be enhanced through a reduction in intracellular cAMP levels [104]. This study highlights that the effect of ADO receptor agonism on cancer cell biology may differ depending upon the distinct downstream signaling evoked in different cell types.

A_2B_R is the predominant receptor expressed in multiple tumor types with several fold higher expression over normal tissue (reviewed in [105]). A number of studies have shown that A_2B_R activation can enhance tumor growth [84,85,106,107,108] and that targeting A_2B_R at the genetic level can reverse this effect [84,106,108]. A_2B_R activation has also been shown to play a role in the migration and metastasis of tumor cells [19,108,109,110,111], due to the ability of A_2B_R to induce the epithelial-mesenchymal transition (EMT) through activation of the ERK1/2 pathway [112,113]. Interestingly, Giacomelli et al. [113] demonstrated that the activation of cAMP (mediated through A_2B_R activation) inhibited the EMT and that effects were more pronounced in the presence of a PKA inhibitor. Thus, distinct environmental cues and/or biased agonism may contribute to the net effect of A_2B_R expression on tumor cells. A_2B_R activation and downstream JunB activation also promoted tumor growth through increased production of VEGF and IL-8, subsequently enhancing angiogenesis [114,115]. Taken together, these observations may explain the negative association between A_2B_R expression and patient outcomes in several cancer types including breast and bladder cancer [108,109].

The A_2A_R is overexpressed in some cancers, such as head and neck squamous cell carcinoma (HNSCC) [28,116], and in several breast and melanoma cell lines, [76,82,117] and has been shown to induce PLC, PKC, AKT, MAPK/ERK, and JNK signaling pathways to promote cell proliferation in vitro [118]. The ERK and JNK pathways have also been shown to be downstream effectors of A_1_R signaling, promoting proliferation and migration of tumors [119]. While antagonists to the A_2A_R and A_1_R have been reported to suppress tumor growth in vitro, a definitive role for suppressing tumor growth in vivo remains to be determined. Koszalka et al. performed a broad study on the in vivo effect of A_1_R, A_2A_R, and A_3_R signaling in B16 melanoma in mice and found that the receptors contribute to the TME by modulating angiogenesis, neovascularization, and infiltration of immunosuppressive tumor associated macrophages (TAMs) [76]. Multiple groups have shown that genetic deletion or blockade of the A_2A_R in mice has potent effects in enhancing anti-tumour immunity and reducing tumor growth and metastasis, indicating that targeting the A_2A_R on immune cells likely has a larger therapeutic effect than targeting this receptor on tumor cells alone [9,110].

## 5. Effect of Adenosine on T Cell Responses

### 5.1. Effector T Cells

Mouse and human T cells express A_2A_R, A_2B_R, and A_3_R [72,120,121]. ADO suppresses the cytokine production and proliferation of both CD8^+^ and CD4^+^ T cells as well as the cytotoxic activity of CD8^+^ T cells [122,123,124,125,126,127,128]. This is largely thought to be mediated through the activation of the A_2A_R as pharmacological blockade of the A_2A_R reduces the effects of ADO analogues and A_2A_R^−/−^ T cells are resistant to these compounds [129,130,131]. In vivo, A_2A_R^−/−^ or CD73^−/−^ T cells exhibit an enhanced inflammatory phenotype [132]. For example, they more profoundly induce colitis [133] and exhibit more potent anti-tumor activity. In the acute setting it is known that the upregulated expression of A_2A_R on T cells is driven through NFAT/NFκB which are activated downstream of the TCR or HIF1α in the context of hypoxia [130,134]. Maximal upregulation is seen rapidly (~4–6 h post activation) suggesting that activated cells within the hypoxic tumor microenvironment may be immediately sensitized to ADO [135]. Rapid upregulation of the A_2A_R is a physiological mechanism to reduce inflammatory tissue damage in multiple organs [136,137], but this pathway is also utilized by tumors to suppress anti-tumor immunity, with increased expression of the A_2A_R being observed on tumor-infiltrating CD8^+^ T cells relative to those isolated from the draining lymph node [131]. Multiple signaling events have been reported downstream of the A_2A_R (Figure 2) but the suppressive effect of A_2A_R activation on effector T cells is predominantly thought to be mediated by increased cAMP levels [60,138,139,140]. However, ADO has also been shown to attenuate signaling through key kinases proximal to the TCR including ZAP-70, ERK1/2, and JunB/AP-1 binding, which has also been associated with suppression of T cell proliferation and reduced calcium-influx into CD8^+^ cells [122,141,142]. A role of A_2A_R mediated inhibition of STAT5 signaling via Shp-2 as well as CpG site demethylation of the IL-2 promoter have also been described in T cells [142,143,144]. Whilst the A_2A_R is thought to be the predominant receptor with regards to suppression of T cell responses, a role for both A_2B_R and A_3_R receptors has also been postulated [121,145]. In contrast to the A_2A_R, some studies have indicated that signalling via the A_3_R enhances T cell function [146,147]. Although not addressed in these studies, one possible mechanism for this would be that A_3_R activation has been reported to decrease intracellular cAMP levels within T cells [72]. A_3_R expression can be increased by cAMP levels mediated by activation of ADO receptors and through T cell activation, which suggests a potential role in negative feedback to the A_2A_R/A_2B_R mediated signaling [148].

### 5.2. Regulatory T Cells

While CD73 is expressed on effector T cells, foxp3+ Tregs are capable of co-expressing both CD39 and CD73, and hence contributing to the generation of ADO in the TME and suppression of anti-tumor immune response [126]. Interestingly the A_2A_R is also highly expressed on foxp3^+^ Tregs and stimulation of foxp3^+^ Tregs with A_2A_R agonists have been reported to result in increased cAMP, proliferation, and induction of foxp3, PD-1, and CTLA-4 expression in vitro [126,149,150,151]. Indeed, it appears that autocrine adenosine signalling via the A_2A_R expressed on Tregs may also be important in their suppressive function [151]. Supporting this rationale, A_2A_R blockade has been reported to reduce the number of tumor-infiltrating Tregs which may contribute to its anti-tumor efficacy in vivo [116].

### 5.3. Regulation of Memory T Cells

Alternative roles of the A_2A_R include reducing T cell motility through the modulation of potassium channels [152], regulation of metabolism, and modulation of memory differentiation through PI3K-AKT pathways [9,122,142,153,154,155]. Although A_2A_R and A_2B_R are more highly expressed in effector memory cells, their activation is associated with the preservation of a naïve/ central memory phenotype [35]. Activation of A_2A_R/A_2B_R has been shown to promote expression of CCR7 [156] and the IL7R [153] whilst reducing FAS/FASL induced cell death [157], highlighting the importance of A_2A_R signaling in maintaining memory and tolerance. Notably, reduced AMPK/mTOR/pS6 signaling has been identified downstream of A_2A_R activation, with a more pronounced phenotype observed in memory T cells [35]. Interestingly, a subset of CD8^+^ T cells co-expressing CD39 and CD103 have been identified as being enriched in tumors and correlate with better overall survival in patients with head and neck cancer [158]. These T cells express a resident memory T cell gene signature and high levels of exhaustion markers, and blockade of CD39/CD73 alone or combination with checkpoint blockade may be a promising therapeutic strategy to improve the anti-tumor potency of these CD103^+^ CD39^+^ T cells [158]. Co-expression of CD38 and CD101 has also been identified as a phenotypic indicator of terminally differentiated PD1^hi^ T cells, which were not responsive to checkpoint blockade [159]. The role of CD39, CD38, and ADO itself in the differentiation of these cells is yet to be demonstrated but these observations suggest that ADO modulates T cell differentiation as well as activity in the TME.

## 6. Effect of Adenosine on Natural Killer or Natural Killer T Cell Responses

Natural Killer (NK) and Natural Killer T cells (NKT) express high levels of the A_2A_R. Indeed, on a per cell basis NK cells express more A_2A_R mRNA than T cells during homeostasis [160,161]. NK cells from wild-type mice, but not A_2A_R^−/−^ mice can be potently suppressed by ADO or specific A_2A_R agonists in terms of their cytotoxic function and cytokine production. Similarly to T cells, expression of the A_2A_R on NK cells is more abundant on the more mature NK subsets (CD56^dim^) as opposed to the immature CD56^high^ precursor population and A_2A_R blockade has been shown to enhance NK cell maturation [58,110,162,163,164]. Similarly to T cells, A_2A_R-mediated enhancement of intracellular cAMP concentration and PKA signaling is thought to be the predominant mechanism by which ADO suppresses NK and NKT cell activity (Figure 2) [163,165]. At the transcriptional level, this may be related to observations that A_2A_R signaling induces foxo-1 expression by inhibiting AKT. Since foxo-1 deficient NK cells have been shown to exhibit an enhanced maturation phenotype, it is possible that the upregulation of foxo-1 by A_2A_R activation limits the maturation and effector functions of NK cells [166]. As with T cells, NK cells also express the A_3_R and their function has been shown to be positively regulated by A_3_R agonists resulting in inhibition of primary and metastatic tumor growth in mouse and human models of colon cancer and melanoma [147,167,168,169].

## 7. Effect of Adenosine on Myeloid Cells

Myeloid cells are highly heterogeneous within the tumor microenvironment and are an important determinant in shaping the anti-tumor immune response. Immature myeloid cells, including myeloid-derived suppressor cells (MDSCs) and “type 2” (M2) macrophages have the potential to suppress T cells through a range of mechanisms, including ADO production via their expression of CD73 and CD39 [170,171]. A_2A_R and A_2B_R are also expressed on a range of myeloid cells and their activation has been shown to modulate the function of monocytes, macrophages, and DCs [172] and promote the expansion of MDSCs [173]. Although the mechanisms by which the expression of the A_2A_R is controlled on myeloid cells remains to be elucidated, the expression of the A_2B_R was shown to be increased following IFNγ stimulation and by hypoxia under the control of the HIF1α transcription factor [174,175]. Furthermore, it has been demonstrated that TGFβ enhances the expression of CD73 and CD39 on MDSCs [176]. Therefore, the CD73: A_2A_R/A_2B_R axis may be an important mechanism of immune modulation by MDSC and TAMs in the context of the TME.

Activation of A_2A_R or A_2B_R has also been shown to enhance the differentiation of alternatively activated macrophages, as shown by the upregulation of several M2 markers including TIMP-1 and arginase 1 [177]. A_2A_R signaling promotes ERK, pAKT, and pSTAT3 in monocytes, leading to enhanced differentiation of M2 macrophages in the TME and increased release of pro-tumorigenic cytokines and factors [155]. Similarly, activation of the A_2B_R and potentially other ADO receptors on DCs was shown to impair DC maturation [178] and induce a tolerogenic phenotype with reduced CD8^+^ T cell priming capacity [179,180]. A_2B_R activation has also been shown to enhance IL-6 production from DCs, and to consequently promote T_H_17 responses [181], the effect of which in cancer is controversial [182]. In the acute setting, activation of A_2A_R or A_2B_R has been shown to suppress pro-inflammatory cytokine production by monocytes [67], macrophages [183], and DCs [184,185,186]. Thus, macrophages stimulated with ADO secrete increased amounts of IL-10 and less IL-12, TNFα, and chemotactic factors. [183,187,188,189]. Using an elegant approach in which the A_2A_R was specifically deleted in myeloid cells (using a Lys-Cre system) Cekic et al. were able to show that the A_2A_R limited anti-tumor immune responses in part through modulation of myeloid cells and subsequent enhancement of anti-tumor T cell responses [190]. Similarly, A_2B_R blockade has been shown to enhance anti-tumor immune responses, partly through a reduction in MDSC differentiation [115,173,191] and partly through an enhancement of the capacity of DCs to evoke anti-tumor T cell responses [172,191].

A_3_R activation has been associated with increased chemotaxis, degranulation and regulation of superoxide production in neutrophils [192,193,194,195], plasmacytoid DCs and monocytes/macrophages [78,196,197,198]. A_3_R expression in intratumoral mast cells is associated with PI3K mediated phosphorylation of ERK/MAPK and AKT resulting in increased release of IL-8 that can promote angiogenesis and EMT in tumor cells [199]. The PI3K and PKC signaling pathway has also been associated with A_3_R in macrophages and has been shown to enhance TNFα release in response to LPS, indicating a possible role of A_3_R in regulating inflammatory cytokine release [78]. The A_3_R may thus be involved in the recruitment, survival, and function of macrophages and mast cells in the TME [76,200].

Taken together, these observations suggest that ADO plays an important role in modulating the activity of tumor-infiltrating myeloid cells by limiting their capacity to evoke anti-tumor immune responses and are potential targets for therapeutic intervention.

## 8. Targeting the Adenosine Pathway in TME to Improve Immunotherapies

Cancer immunotherapies have been hailed as the fourth pillar of cancer treatment and its great success in the clinic has led to the Nobel prize recently being awarded to James Allison and Tasuku Honjo for recognition of their discovery of the checkpoint molecules CTLA-4 and PD-1 respectively. Checkpoint receptor blockade can lead to durable responses in a range of cancers, however not all patients respond to treatment, highlighting the need for further research to understand tumor evasion mechanisms and identify other targets that can overcome these ‘brakes’ on the immune response. As described in previous sections, ADO is known to act through the A_2A_R to negatively regulate T and NK cell responses in the TME and naturally targeting this pathway may further enhance the efficacy of immunotherapies in the clinic.

Seminal studies from the Sitkovsky and Powell groups demonstrated that the genetic deletion of the A_2A_R can potently enhance anti-tumor responses in mice due to the activation or enhancement of T cells in the TME [9,201]. This has led to attempts from multiple groups to therapeutically target this pathway. The main targets for this pathway are the ectoenzymes CD73, CD39, and CD38 which promote ADO formation, and also the downstream A_2A_R. The potential of targeting CD73 was shown by the observation that reducing the expression of CD73 in the ID8-ova ovarian tumor cell line increased their susceptibility to T cell mediated killing in vitro and in vivo. Similarly, anti-CD73 antibodies were shown to reduce tumor growth and metastasis through activation of NK and T cell responses [19,20,125]. These effects in the primary tumor setting were shown to be T cell dependent, with the efficacy of anti-CD73 being lost in mice lacking T cells. Interestingly, the dual blockade of CD73 and the A_2A_R has been reported to elicit improved anti-tumor effects. This may be because tumors increased their expression of CD73 in A_2A_R^−/−^ mice and in response to A_2A_R inhibition, potentially highlighting the importance of CD73 as an escape mechanism to anti-tumor T cell responses [202,203]. Considering the immunosuppressive role of A_2A_R on T and NK cells, rational combination strategies targeting the ADO pathway with checkpoint inhibitors and adoptive cell therapies (ACT) have the potential to synergistically enhance anti-tumour immune cell function. CD73, and more recently CD38, expression on tumor cells have been shown to confer resistance to anti-PD-1, as activation of T cells and PD-1 blockade upregulates the expression of A_2A_R, making these cells more susceptible to adenosine-mediated suppression [131]. Hence inhibition of CD73, A_2A_R or CD38 in combination with anti-PD-1, has been shown to elicit enhanced anti-tumour T cell responses mediated by enhanced IFNγ and Granzyme B expression by CD8^+^ T cells [131,204,205]. Another study demonstrated that A_2A_R^−/−^ T cells could better penetrate hypoxic tumors, which could then be targeted with dual checkpoint blockade to further enhance anti-tumor function [13]. Although the majority of combination approaches have focused on the ability of ADO targeting to enhance T cell responses, other immune subsets within the TME can also be enhanced by targeting the ADO axis including NK cells [110] and therefore future combination approaches may explore reagents which stimulate other immune cell subsets modulated by ADO. Similarly, this strategy can potentially be applied to ACT and chimeric antigen receptor T cell therapy (CAR-T), whereby tumor specific autologous TILs are extracted, expanded ex-vivo with or without modifications before reinfusion back into the patient in large numbers. Indeed, it has been demonstrated that targeting the A_2A_R using genetic or pharmacological approaches could enhance the efficacy of CAR T cells or conventional ACT [125,128,206,207].

Chemotherapy remains a major front-line option for many cancer patients, and it is now well accepted that the immune response is a major determinant governing the response to treatment. Therefore, there is huge potential for synergy by combining A_2A_R antagonists or anti-CD73/ anti-CD39 with chemotherapy. Evidence suggests that the ADO axis is enhanced following chemotherapy and modulates the therapeutic response. For example, an anthracycline-based chemotherapeutic agent, doxorubicin (DOX), was shown to induce CD73 expression on tumor cells [24]. Another recent study demonstrated an increase in CD73 and PD-L1 expression by DOX, gemcitabine, or paclitaxel chemotherapy mediated by increased expression of HIF1α and HIF2α [208]. As such, blocking CD73 or A_2A_R directly or through inhibition of HIF and subsequent expression of these receptors can significantly enhance anthracycline-based chemotherapy and may be viable as a future combinatorial strategy in the clinic [24].

Multiple small molecule inhibitors and antagonistic antibodies against these targets have been developed and show promising therapeutic efficacy against different solid tumors in clinical trials (Table 1). A_2A_R antagonists SYN115 and Istradefylline have been trialed to improve motor function in patients with Parkinson’s disease and are well tolerated with the most common adverse effects being dyskinesia, nausea, and dizziness [209,210]. Other A_2A_R antagonists are being trialed to treat cancer, which include CPI-444 (NCT02655822, NCT03454451), PBF509 (NCT02403193), NIR178 (NCT03207867), and AZD4635 (NCT02740985, NCT03381274). CPI-444 in combination with anti-PD-1 and anti-CTLA4 was highly effective in promoting CD8^+^ T cell responses and eliminating tumors in a preclinical model [211]. Together these preclinical findings have led to the A_2A_R antagonists PBF-509 (NCT02403193) and CPI-444 (NCT03337698) being trialed for safety and efficacy against HNSCC, NSCLC, melanoma, renal cell cancer, triple-negative breast cancer (TNBC), colorectal, bladder, and prostate cancers [212]. Antibodies targeting CD73 have also progressed to early stage clinical trials, both alone and in combination with anti-PD-1 or antagonists of the A_2A_R (NCT02503774, NCT03267589, NCT03616886, NCT03381274, NCT03454451, NCT03549000, NCT02754141). Similarly, two antibodies targeting CD38 have been developed, Daratumumab (NCT02944565) and isatuximab (NCT01084252), and have been trialed in multiple myeloma. Based on the observations aforementioned that CD38 may be a key mediator of resistance to checkpoint blockade, these potential combinations in the solid tumor setting warrant investigation.

Although the majority of clinical development targeting ADO receptors has focused on the A_2A_R, reagents targeting the alternative ADO receptors are also of interest. Blocking the A_2B_R receptor has been shown to inhibit melanoma, prostate, and breast cancer growth in mice and to reduce tumor metastasis [106,109,213]. In the clinic, the A_2B_R antagonist PBF1129 is currently being trialed in patients with non-small cell lung cancer (NSCLC) (NCT03274479). A_2B_R targeting has a potential dual benefit of limiting tumor cell growth and metastasis as well as enhancing anti-tumor immune responses through modulation of both lymphoid and myeloid subsets. Interestingly, dual A_2B_R/A_2A_R antagonists have now been developed with the potential to further enhance anti-tumor effects. The A_1_R and A_3_R are other potential ADO receptors which could be targeted, but while the A_1_R antagonist, DPCPX has been shown preclinically to inhibit tumor cell proliferation, migration, and promote apoptosis, no drugs targeting the A_1_R are currently undergoing clinical trials. Currently there is only one A_3_R agonist, CF102, undergoing clinical trials to treat hepatocellular carcinoma (NCT00790218, NCT02128958). Further preclinical studies are required to elucidate the potential for combining A_1_R or A_3_R agonists/antagonists with chemo- or immunotherapies.

## 9. Conclusions

Escaping immune destruction is one of the hallmarks of cancer and overcoming suppressive mechanisms in the tumor niche is a major focus for enhancing immunotherapies. ADO, a metabolite of the ubiquitous energy molecule ATP, is one such mechanism shown to have broad immunosuppressive activities. Reflecting the important role of ADO in the tumor context, CD39, CD73, and CD38 ectoenzymes have been identified as potential biomarkers for clinical outcomes in chemo- and immune-therapies and to identify immune subsets that may be responsible for immunosuppression or are in an exhausted state. Moreover, clinical grade reagents targeting these ectoenzymes, along with the downstream ADO receptors A_2A_R and A_2B_R have now entered clinical trials alone or in combination with other immunotherapy approaches. Several challenges remain in targeting the adenosine immunosuppressive pathway. Firstly, although mice deficient for members of this pathway, including CD73, CD39, and A_2A_R, exhibit mild autoimmune phenotypes compared to mice lacking other immune checkpoints such as PD-1 and CTLA-4, the potential remains for unwanted autoimmune/inflammatory side-effects. Moreover, given the ubiquitous expression of adenosine receptors in many cell subtypes the potential for on-target side effects exists. Despite the development of clinical reagents, the underlying mechanisms of these therapies are relatively unknown, particularly in terms of the signaling events mediated by ADO in distinct immune cell subtypes. Understanding of this is hampered by a lack of studies investigating ADO receptor signaling events in primary immune cells. This is likely to be a focus of future studies given that the outcome of ADO receptor activation is dependent on the cell-specific expression of signaling proteins including receptor subtypes as well as downstream signaling molecules. We have summarized the signaling events that have been verified in each of the respective primary immune subsets in Figure 2. In addition, much of the information about ADO receptor signaling has been derived using pharmacological agents which are considered to selectively activate/block certain ADO receptor subtypes. The conclusions from studies like this are potentially confounded as other studies have shown that ADO receptor ligands can also mediate ADO receptor independent effects [214]. Thus, validation of signaling events should be ideally confirmed at the genetic level in primary immune cells. Nevertheless, the overall evidence speaks for the high therapeutic potential of targeting the ADO axis, and further studies which provide further mechanistic insight into this are likely to present the potential for more rational therapeutic combinations.

The inflammatory and hypoxic tumour microenvironment drives the expression of the ectoenzymes CD39, CD73, and CD38 on tumors, cancer associated fibroblasts (CAFs), regulatory T cells (Treg), and myeloid derived suppressor cells (MDSCs), which catalyze the conversion of ATP and NAD^+^ into extracellular adenosine. Adenosine is rapidly converted to inosine by adenosine deaminase (ADA) expressed on the cell surface. Adenosine and inosine signal through multiple adenosine receptor subtypes (A_1_R, A_2A_R, A_2B_R, and A_3R_), with A_2A_R and A_2B_R playing a predominant role in the suppression of anti-tumor immune cell responses. The A_2A_R is highly expressed on T effectors (Teff), natural killers (NK), and Tregs relative to the other receptor subtypes. Adenosine modulates multiple functions of tumor infiltrating Teff, Tregs, NKs, MDSCs, tumor associated macrophages (TAM), and dendritic cells (DCs) including differentiation, proliferation, cytokine production, and cytotoxic function.

A summary of the identified signaling pathways downstream of the A_1_R, A_2A_R, A_2B_R, and A_3_R in T effectors (Teff), Natural Killer (NK), tumor associated macrophages (TAM), and the tumor cells themselves is presented. The A_1_R and A_3_R are coupled to the Gαi/o subunit and inhibit adenylate cyclase and cyclic adenosine monophosphate (cAMP) production while the A_2A_R and A_2B_R are coupled to the Gαs which promote cAMP accumulation. Protein kinase A (PKA), extracellular signal-regulated kinases 1 and 2 (ERK1/2), protein kinase B (AKT), phospholipase C (PLC), phosphatidylinositide 3-kinase (PI3K), protein kinase C (PKC), c-Jun N-terminal kinases (JNK), SHP-2, and the mechanistic target of rapamycic (mTOR) can signal downstream of the adenosine receptors to regulate cell specific responses in the TME.

## Figures and Tables

**Figure 1 ijms-19-03837-f001:**
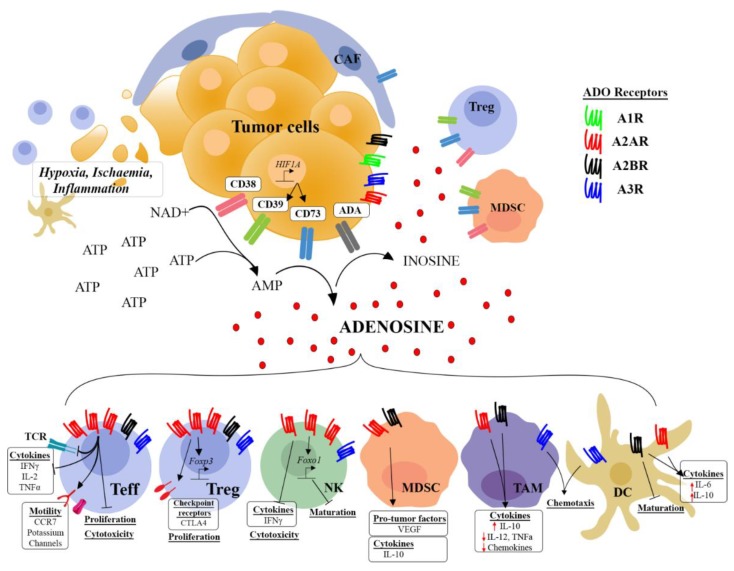
Generation of adenosine in the tumor microenvironment leads to the suppression of multiple immune subsets. Arrows indicate increased expression or activation. T bars indicate inhibition or reduced activity.

**Figure 2 ijms-19-03837-f002:**
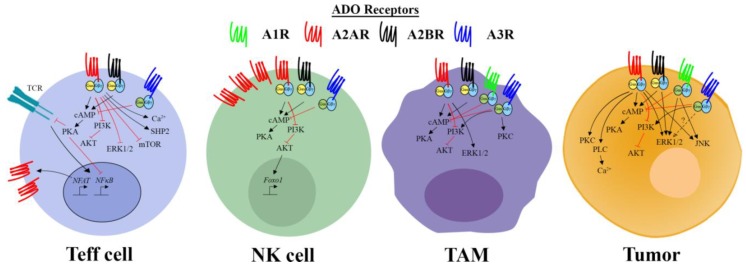
Expression of adenosine receptors and their downstream signalling pathways within various immune cell subsets and tumor cells in the context of the TME. Arrows indicate increased expression or activation. T bars indicate inhibition or reduced activity.

**Table 1 ijms-19-03837-t001:** Ongoing or upcoming clinical trials targeting the adenosine pathway.

Target	Drug	Company	Clinical Trial Number	Study Phase	Cancer Type	Combination
A_2B_R (Antagonist)	PBF-1129	Palobiofarma	NCT03274479	I	NSCLC	
A_2A_R (Antagonist)	CPI-444	Corvus	NCT02655822	I/Ib	Solid cancers	atezolizumab
			NCT03337698	I/II	Carcinoma, NSCLC	Multiple drug combinations
	PBF-509	Palobiofarma	NCT02403193	I/II	NSCLC	PDR-001 (αPD-1)
	NIR-178	Novartis	NCT03207867	II	Solid cancers and DLBCL	PDR-001 (αPD-1)
	AZD-4635	Heptares	NCT02740985	I	Solid cancers	Durvalumab (αPD-L1)
A_3_R (Agonist)	CF-102	CanFite BioPharma	NCT02128958	II	Hepatocellular carcinoma	
CD73	MEDI-9447	MedImmune	NCT02503774	I	Solid cancers	Durvalumab (αPD-L1)
			NCT03267589	II	Ovarian cancer	Durvalumab (αPD-L1), Tremelilumab (αCTLA4), MEDI 0562 (αOX-40)
			NCT03616886	I/II	TNBC	Durvalumab (αPD-L1), Paclitaxel, Carboplatin
			NCT03381274	I/II	Carcinoma, NSCLC	Durvalumab (αPD-L1), Osimertinib
	CPI-006	Corvus	NCT03454451	I	Solid cancers	Pembrolizumab (αPD-L1), CPI-444 (A_2A_Ri)
	NZV-930	Norvatis	NCT03549000	I	Solid cancers	PDR001 (αPD-1), PBF-509 (A_2A_Ri)
	BMS-986179	Bristol-Meyers-Squibb	NCT02754141	I/II	Solid cancers	Nivolumab, rHuPH20

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
