# Peer review of "Targeting Adenosine Receptor Signaling in Cancer Immunotherapy"

_ijms, 2018, doi:10.3390/ijms19123837_

Round 1
Reviewer 1 Report
The review summarizes the adenosine pathways and how those pathways are targeted in tumor immunotherapy. Specifically, it focuses on the expression patterns and action mechanism of adenosine receptors and ectoenzymes mediating adenosine production in immune cells of tumor microenvironment and tumor cells. The last section discusses the ways to incorporate adenosine pathway regulation in cancer immunotherapy.
General comments
Adenosine pathways in tumor immunotherapy is an extensive topic as it has an impact on so many components of the immune system. Therefore, as this review is very comprehensive, it will be easier to read if each the main sections was subdivided to smaller subsections. For example, Section 5, “Effect of adenosine on T cell responses”, can be organized in a way that separates between the regulation of CD8 cells, T reg cells, and a memory component. A better organization of the content of each section can significantly improve the overall readability of this review article.
Specific comments
There are some spelling errors thought the manuscript. For example, Figure 1 legend: “tumor”; Line 231: “utilized.”
The discussion section should also include the challenges of targeting the adenosine pathway in cancer immunotherapy. One such challenge is that widely spread expression of adenosine receptors can potentially result in severe adverse reactions.
Author Response
We thank the reviewer for this suggestion and we have now split up the section as suggested. The new section can be found at line 224 of the revised manuscript.
Specific comments
1) There are some spelling errors thought the manuscript. For example, Figure 1 legend: “tumor”; Line 231: “utilized.”
Author Response:
We thank the reviewer for pointing out these grammatical errors which have now been corrected.
2) The discussion section should also include the challenges of targeting the adenosine pathway in cancer immunotherapy. One such challenge is that widely spread expression of adenosine receptors can potentially result in severe adverse reactions.
Author Response:
We agree this information would enhance the comprehensiveness of the review. We have inserted the following text at line 450 of the revised manuscript.
“Several challenges remain in targeting the adenosine immunosuppressive pathway. Firstly, although mice deficient for members of this pathway including CD73, CD39 and A2AR exhibit mild autoimmune phenotypes compared to mice lacking other immune checkpoints such as PD-1 and CTLA-4, the potential remains for unwanted autoimmune/ inflammatory side-effects. Moreover, given the ubiquitous expression of adenosine receptors in many cell subtypes the potential for on-target side effects exists.”
Reviewer 2 Report
The authors provide a comprehensive review on adenosine signaling, as potential therapeutic target in cancer immunotherapy. The manuscript is well written and interesting to the readers not only in basic scientific research, but also in pharmaceutical industry. Updated references are cited.
The paper is acceptable for publication in its present form.
Author Response
We thank the reviewer for their review of our manuscript.